# GC-MS Profiling of Ethanol-Extracted Polyherbal Compounds from Medicinal Plant (*Citrullus colocynthis*, *Curcuma longa*, and *Myristica fragrans*): In Silico and Analytical Insights into Diabetic Neuropathy Therapy via Targeting the Aldose Reductase

**DOI:** 10.3390/cimb47020075

**Published:** 2025-01-23

**Authors:** Mohd Adnan Kausar, Sadaf Anwar, Halima Mustafa Elagib, Kehkashan Parveen, Malik Asif Hussain, Mohammad Zeeshan Najm, Abhinav Nair, Subhabrata Kar

**Affiliations:** 1Department of Biochemistry, College of Medicine, University of Ha’il, Al Khitah Street, Hail 55476, Saudi Arabia; sa.anwar@uoh.edu.sa; 2Department of Pharmacology, College of Medicine, University of Ha’il, Hail 55476, Saudi Arabia; h.elagib@uoh.edu.sa; 3Interdisciplinary Biotechnology Unit, Aligarh Muslim University, Aligarh 202002, Uttar Pradesh, India; kehkashan.parveen@gmail.com; 4Department of Pathology, College of Medicine, University of Ha’il, Hail 55476, Saudi Arabia; mh.hussain@uoh.edu.sa; 5School of Biosciences, Apeejay Stya University, Gurugram, Sohna 122103, Haryana, India; biotechzeeshan@gmail.com; 6Department of Life Science, Sharda School of Basic Sciences and Research, Sharda University, Greater Noida 201310, Uttar Pradesh, India; abhinav.nair@sharda.ac.in (A.N.); kar.subha@gmail.com (S.K.)

**Keywords:** diabetic neuropathy, pharmacokinetics, amputation, diabetic peripheral aldol reductase, docking

## Abstract

Diabetic neuropathy is one of the severe complications of diabetes, which affects the quality of life in a patient and increases the risk of amputations and chronic wounds. Current therapeutic approaches are symptomatically oriented, focusing on comfort and non-inflammatory aspects without addressing the mechanism or molecular target of the disease. The present study investigates the therapeutic effects of an ethanolic polyherbal extract from *Citrullus colocynthis* (Bitter Apple), *Curcuma longa* (Turmeric), and *Myristica fragrans* (Nutmeg) using advanced in silico and analytical methods. According to the findings, PHE showed the presence of a total of 39 bioactive compounds in GC–MS analysis, which include alcohols, fatty acids, terpenoids, esters, neolignans, phenylpropanoids, and steroids. Three of the compounds—-4-isopropyl-1,6-dimethyl-1,2,3,4-tetrahydronaphthalene (−11.4 kcal/mol), (1S,2R)-2-(4-allyl-2,6-dimethoxyphenoxy)-1-(3,4,5-trimethoxyphenyl)-1-propanol (−9.8 kcal/mol) and (S)-5-Allyl-2-((1-(3,4-dimethoxyphenyl)propan-2-yl)oxy)-1,3-dimethoxybenzene (−10.3 kcal/mol)—followed the Lipinski rule and showed the binding affinity with aldol reductase. Docking experiments showed that compound 4-isopropyl-1,6-dimethyl-1,2,3,4-tetrahydronaphthalene (−11.4 kcal/mol) has high-affinity binding to aldose reductase, an enzyme involved in diabetic neuropathy pathophysiology, whereas molecular dynamics simulations show long-range persistence of the interaction of (S)-5-Allyl-2-((1-(3,4-dimethoxyphenyl)propan-2-yl)oxy)-1,3-dimethoxybenzene with aldol reductase in physiological conditions. Therefore, this combination of herbal therapy and advanced computational/analytical techniques could be leading towards innovative, multi-targeted therapies against diabetic neuropathy. Nevertheless, further studies in vivo are required to confirm the efficacy, safety, and pharmacokinetics of the PHE in biological systems.

## 1. Introduction

A chronic consequence of diabetes mellitus is a diabetic neuropathy that increases the risk of potentially lethal cardiovascular, metabolic, and infectious events in diabetic patients and affects many people with diabetes globally [1]. This disorder causes damage to the nerves, which leads to severe lower extremity pain, paresthesia weakness, and loss of coordination [2]. These symptoms negatively impact quality of life and daily activities and predispose individuals to injuries or infections, leading to amputations and, therefore, contributing to physical disability [2,3]. The elderly who suffer from diabetes have increased risks of falls associated with sensory deficits, muscle weakness, and poor coordination resulting in fractures and further disability [1]. Diabetic neuropathy patients suffer a great challenge of foot ulceration and inflammation [4,5]. The pathogenesis of diabetic complications is indeed multifaceted, and the polyol pathway plays a significant role in mediating osmotic and metabolic disruption in various tissues [6]. Patients with chronic diabetic foot ulceration face greater risk of mortality [7]. A natural-based drug with minimal side effects is needed to treat chronic foot ulcers to prevent the amputation of lower extremities [8]. Targeting the protein involved in hyperglycemic effects is important for managing diabetic neuropathy. Aldose reductase (AR) is a nicotinamide adenine dinucleotide (NAD)-dependent, rate-limiting, cytoplasmic enzyme that plays a key role in the polyol pathway, which converts glucose into sorbitol [9]. The accumulation of sorbitol within nerve cells disrupts the transport of electrons in mitochondria that can lead to oxidative stress, damaging nerve fibers and causing severe pain symptoms (Figure 1) [10,11]. Recent studies show that aldol reductase is identified as a potential target for drugs or bioactive compounds for treatment of both Cancer and Diabetic Neuropathy [12]. Aldol reductase enzyme is highly expressed in animal and human tissues including testis, liver, kidney, skeletal, cardiac, and brain [13]. Involvement of AR has been studied in a variety of diseases including Alcohol Liver Diseases (ALD) and other kidney and lung diseases [12,14]. Suppressing the activity of aldose reductase is considered beneficial in treating diabetic neuropathy and tumor progressions [10]. Several aldose reductase inhibitors, such as zenerastat, tolrestat, and epalrestat, have been developed, but they were discontinued due to unclear clinical efficacy, and provide only partial relief and serious adverse effects on body functions [15]. As a result, scientists are exploring alternative therapeutic approaches. Medicinal plants contain secondary metabolites such as flavonoids, phenolic acids, tannins, terpenoids, and alkaloids, which possess antioxidant, anti-cancer, anti-inflammatory, and vascular protective functions [16,17]. These compounds have shown promising results in treating chronic conditions like diabetes [18,19]. The development of polyherbal extract-based formulations is one promising approach. For the extraction of volatile organic compounds, Carboxen, PDMS fibers, a cold extraction method using ethanol was more utilized than other extraction solvents such as hexane, ether, and chloroform [20]. These formulations incorporate various medicinal plants that may exert synergistic effects, potentially enhancing therapeutic responses. The polyherbal formulation can target multiple pathways simultaneously, providing an effective remedy. The extract of the chosen plant (*Citrullus colocynthis* (Bitter Apple), *Curcuma longa* (Turmeric), *Myristica fragrans*, (Nutmeg; Jaiphal), *Myristica fragrans* (Nutmeg; Jawitri)) contains bioactive compounds that have immense anti-diabetic and anti-cancer potential that may reduce oxidative stress, inflammation, and discomfort associated with diabetic neuropathy. Modern techniques, such as GC-MS profiling, can identify and quantify these bioactive components, offering insight into the composition of the extract and its potential efficacy [21].

In this current study, we aim to evaluate a selected polyherbal extract of medicinal plants (*Citrullus colocynthis* (Bitter Apple), *Curcuma longa* (Turmeric), *Myristica fragrans* (Nutmeg; Jaiphal), *Myristica fragrans* (Nutmeg; Jawitri)) for its therapeutic activity against neuropathy and investigate the key bioactive compounds (**39**) within the extract using GC-MS profiling. The selected compounds have previously been identified in extracts of these medicinal plants [22,23,24]; however, their interaction with aldose reductase as a potential target for diabetes has not yet been reported. Subsequent molecular dynamics (MD) simulations provide insights into how the selected compounds interact with the aldol reductase protein. This research may open new avenues for effective treatment of diabetic neuropathy with traditional herbal medicine supported by modern scientific techniques. This multidisciplinary approach not only lays the groundwork for future research and development in the field of herbal medicine but also highlights the integration of traditional knowledge with contemporary science.

## 2. Material and Methods

Plant materials were purchased from a local market in Delhi, India, and they may be utilized by locals in conventional medicine.

### 2.1. Preparation of Polyherbal Extract

Plant materials (Table 1) were mixed in equal proportions 1:1:1:1 ratio and 25 gm each by weight to make a polyherbal extract (PHE). The respective parts of the plant material were dried in a shed and coarsely powdered for extraction. For 3 days, Soxhlet equipment was used to extract the coarse powder using ethanol at 60–70 °C. After cooling, the extract was filtered through Whatman filter paper to remove any leftovers. After concentration under low pressure in a rotary evaporator, the extracts were dried to produce a powder. The crude extract, PHE, was stored in an airtight bottle at room temperature and sent for GC-MS investigation.

### 2.2. GC-MS Analysis of PHE

PHE was subjected to GC-MS analysis using the Shimadzu GCMS-QP-2010 Plus instrument. The apparatus consists of the Rtx^®^-5MS low-bleed column measuring 30 mm × 0.25 mm with ID × 0.25 µm films. Utilizing a carrier gas (helium) with 1.0 mL/min flow rate, the column was set to operate with the following conditions: the oven was set to operate between 140 and 280 °C for 56 min, with 5 °C per minute increment and a hold period of 2 min. At 260 °C, the injector temperature was kept constant. An injection of 0.3 µL of sample was used at 85.2 kPa of pressure. With a purge flow of 3.0 mL/min and a linear velocity of 41.6 cm s^−1^, the column flow was 1.21 mL/min, and the total flow was 16.3 mL/min. A temperature of 230 °C was used to balance the ion split ratio. The GC-MS data were interpreted using the NIST and Wiley database libraries [25].

### 2.3. Computational Study

#### 2.3.1. Molecular Docking

For molecular docking, virtual high-throughput screening and visualization utilized several computational tools, including AutoDock Vina v.1.2.x [26], PyMOL 2.5.2, Biovia Discovery Studio 2024 [27], Open Babel 2.3 GUI, and SMILES Translator. The protein and ligand information was retrieved and the experimental data was assessed using online resources such as the Protein Data Bank (PDB), PubChem, NCBI, and online accessible Swiss ADME web tool.

#### 2.3.2. Protein Preparation

The three-dimensional crystal structure of Aldol reductase was obtained from the Protein Data Bank using the PDB ID 2R24 with a resolution of 1.75 Å. The protein structure was then cleaned by removing any extra chains or solvent molecules using PyMOL 2.5.2 software. Finally, the structure was saved in PDBQT format to facilitate molecular docking.

#### 2.3.3. Ligand Preparation

The study conducted a screening of various compounds by creating a library of bioactive compounds found in the plant extract. For virtual high-throughput screening (vHTS), these ligands were downloaded from PubChem in 3-D-SDF or MOL format, along with their specific compounds. The screening library includes compound structures in 3D format in PDBQT form, which were subsequently utilized for molecular docking and pharmacological assessment using Swiss ADME [28].

#### 2.3.4. Docking

Discovery Studio was utilized to add hydrogen atoms and charge to both the ligand and protein structures. Compounds that met Lipinski’s criteria [29,30] were selected based on all pharmacokinetic parameters. In silico docking was performed using a specific grid size for the protein, with dimensions of X = 60, Y = 50, and Z = 60 and a spacing of 1 Å. Docking was executed using Vina v.1.2.x software with the appropriate commands. High-binding affinity ligands were identified and visualized in Biovia Discovery Studio 2024 and PyMOL 2.5.2 to analyze the hydrophobic and hydrogen interactions between the active site residues of the protein and the ligands.

#### 2.3.5. Molecular Dynamics (MD)

Molecular dynamics (MD) simulations were used to investigate the interactions of the best protein–ligand complex obtained after molecular docking studies [31], i.e., between 2R24 (Human Aldose Reductase) and (S)-5-Allyl-2-((1-(3,4-dimethoxyphenyl)propan-2-yl)oxy)-1,3-dimethoxybenzene. MD simulations were carried out using GROMACS 2023.2 for 200 ns. The CHARMM36 forcefield was utilized to create a dodecahedron box to enable the solvation of the protein–ligand complexes within the box. The water model used in the MD simulation was a TIP3P water model [32,33]. Periodic boundaries were used. The ionization state of protein residues was kept at pH 7. Optimization of initial geometry was performed using a steep descent algorithm with 5000 iterations. During the equilibration stage, the V-rescale thermostat was used for the temperature coupling method within a constant NVT (number of particles, volume, and temperature) ensemble. Ion charge neutralization was achieved by adding sodium ions. NPT (number of particles, pressure, and temperature) ensemble used V-rescale as the thermostat and C-rescale as the barostat [34]. The final MD production run phase of the MD simulation ran for 200 ns. This aided in obtaining detailed aspects of molecular dynamics and interactions between the protein–ligand complexes. Following the production phase of the simulation, the trajectory analysis was conducted. The root mean square deviation (RMSD), root mean square fluctuation (RMSF) analysis was performed. With the help of these analyses, dynamic structural changes as well as the conformational stability of the protein–ligand complex vis à vis the unbound protein could be discerned.

## 3. Results

### 3.1. Gas Chromatography and Mass Spectrometry Analysis

The GC-MS analysis of PHE revealed 39 components with different *m*/*z* ratios and retention times. The bioactive components were listed in Table 2 along with their retention time (RT), concentration (area of peak percentage), molecular weight, molecular formula, and chemical nature. The chromatogram of all the components combined is shown in Figure 2. Appendix A displayed each different component’s mass spectrum that was derived from PHE. The major identified compounds were tetradecanoic acid (20.38%), myristic acid TMS derivative (17.95%), tetracosapentaene, 2,6,10,15,19,23-hexamethyl (8.75), followed by benzene, 1,2,3-trimethoxy-5-(2-propenyl)-(5.53), 1,3-benzodioxole, 4-methoxy-6-(2-propenyl)-(4.45%), dodecanoic acid, TMS derivative (3.84%), (1S,2R)-2-(4-allyl-2,6-Dimethoxyphenoxy)-1-(3,4,5-trimethoxyphenyl) propan-1-ol-rel- (3.80%), (1S,2R)-2-(4-Allyl-2,6-dimethoxyphenoxy)-1-(3,4-dimethoxyphenyl)propyl acetate (3.27%), Cholesta-5,20-dien-3-ol, (3.beta.)-(2.76%), adipic acid, 2,4-dimethylpent-3-yl eicosyl ester (2.70%), p-menth-8-en-3-ol, acetate (2.54%) and hexadecanoic acid, and 2-hydroxy-1,3-propanediyl ester (2.37%; Table 2). They fall into a variety of chemical categories, and the majority are considered to have high levels of bioactivity (antioxidant, anti-inflammatory, etc.). PHE contained recognized bioactive substances such as phenylpropanoids, fatty acids, terpenoids, esters, neolignan, steroids, and alcohols according to the GC-MS analysis. Phenylpropanoids constitute the major part present in GC-MS analysis of polyherbal extract. These have multifaceted effects including antioxidant, antidiabetic, antimicrobial, and anti-inflammatory properties. Others considerable compounds such as fatty acid methyl esters, terpenoids, and neolignanas were reported to consist of antibacterial, anti-inflammatory, antioxidant, and antifungal activities.

### 3.2. Molecular Docking

With the help of Swiss ADME, 39 chosen compounds were screened using GC-MS with an emphasis on their physical and chemical characteristics (Appendix A). Eleven of these compounds met the Lipinski criteria and were selected for docking investigation, as shown in Table 3. From the selected compounds, three exhibited high binding affinities against the aldol reductase protein: 4-isopropyl-1,6-dimethyl-1,2,3,4-tetrahydronaphthalene (−11.4 kcal/mol), (1S,2R)-2-(4-allyl-2,6-dimethoxyphenoxy)-1-(3,4,5-trimethoxyphenyl)-1-propanol (−9.8 kcal/mol), and (S)-5-Allyl-2-((1-(3,4-dimethoxyphenyl)propan-2-yl)oxy)-1,3-dimethoxybenzene (−10.3 kcal/mol). The 3D interactions of these compounds with aldol reductase are shown in Figure 3A–C. The compound 4-isopropyl-1,6-dimethyl-1,2,3,4-tetrahydronaphthalene (−11.4 kcal/mol) interacts with the Thr135 residue of aldol reductase, forming one conventional hydrogen bond. Less hydrogen bond contacts (less than 3.0 Å) occur between the ligand and aldol reductase; instead, the majority of interactions are hydrophobic. Together with hydrophobic interactions with the Val130, Val131, and Lys307 residues of aldol reductase, the compound (1S,2R)-2-(4-allyl-2,6-dimethoxyphenoxy)-1-(3,4,5-trimethoxyphenyl)-1-propanol also exhibited one hydrogen bond contact with the Ser133 residue. With the Asp134 residue of aldol reductase, on the other hand, the ligand (S)-5-Allyl-2-((1-(3,4-dimethoxyphenyl)propan-2-yl)oxy)-1,3-dimethoxybenzene has significant hydrophobic contacts but no hydrogen bond interactions.

### 3.3. Molecular Dynamic Simulation

#### 3.3.1. RMSD

The MD analysis used RMSD and RMSF to study the conformational stability of the protein–ligand complex during a 200 ns-long simulation [35]. The MD simulation between Human Aldose Reductase and the top ligand compound (S)-5-Allyl-2-((1-(3,4-dimethoxyphenyl)propan-2-yl)oxy)-1,3-dimethoxybenzene and the subsequent trajectory analysis using RMSD (showing changes in the atoms or residues of the protein as well as the ligand over simulation time) revealed that the protein–ligand complex was more stable compared to the unbound protein form as shown in Figure 4A. RMSD values for the unbound protein was higher compared to the protein–ligand complex in the initial 40–50 ns. Between 50 and 150 ns, both the unbound protein and the protein–ligand complex overall had similar RMSD values (0.1 to 0.25). However, after 150 ns the protein–ligand complex stabilized with downward RMSD values reaching around 0.15 at 200 ns. On the other hand, the unbound protein continued to have increased RMSD values after 150 ns and reached close to 0.25 at 200 ns.

#### 3.3.2. RMSF

RMSF, which is used to reveal the protein residues showing fluctuations during the MD simulation, showed overall low RMSF values. Except for the terminal region of the ligand-bound protein, RMSF values were below 0.6 Å, and the terminal region showed a slightly higher (>0.6 Å) RMSF value. In the overall duration of the simulation, no residue of the protein-bound ligand exceeded 0.7 Å, as depicted in Figure 4B, therefore, indicating stable interactions between the protein and its ligand. Notably, most residues show RMSF values less than 0.3 Å.

## 4. Discussion

Diabetic neuropathy (DN) is the condition caused by diabetes, which causes damage to both motor and sensory neurons. It results in several physical impairments [2]. Current treatment methods available are symptomatic rather than curative for the root cause of the disease. Lately, the neuroprotective, anti-inflammatory, and antioxidant properties of phytochemicals have brought much attention towards these plant-derived substances. These phytochemicals have exhibited potential therapeutic use for the treatment of DN and its physical disabilities. Among various phytochemicals identified for the targeted pathogenic pathways of DN, several are flavonoids, alkaloids, terpenoids, and phenolic compounds [17]. Aldose reductase is an enzyme very important in the polyol pathway that can enhance the oxidative stress in diabetic neuropathy (DN). The compounds such as berberine, ellagic acid, and rutin have been reported to be inhibitors of this enzyme [12,36].

In the present study, we evaluated the plant extracts from nutmeg, turmeric, and bitter apple in search of the bioactive potential of phytocompounds that might inhibit the polyol pathway (Table 1). The study objective was to target the aldose reductase enzyme in order to alleviate oxidative stress and the manifestations of hyperglycemia that eventually lead to the physical disabilities associated with diabetic neuropathy. The PHE were subjected to GC-MS analysis and the different mass spectrum and retention time of each bioactive compound mentioned in Appendix A was observed. The identified bioactive compounds are majorly phenylpropanoids, fatty acids, terpenoids, esters, neolignan, steroids, and alcohols.

Online accessible SwissADME web tool was used to assess and predict the physical and chemical properties of 39 bioactive substances. Eleven of these compounds were chosen for further molecular docking research after they fulfilled Lipinski’s rule of five (Table 3). With aldol reductase, three compounds showed the best high-binding affinities. The isolated compounds have already been reported in these medicinal plants in previous studies [22,23,24,37]. With a binding affinity of −11.4 kcal/mol, the first compound, 4-isopropyl-1,6-dimethyl-1,2,3,4-tetrahydronaphthalene, showed the greatest affinity. Prior research demonstrated this compound significance in myocardial infraction and its cardioprotective function in mice with diabetes [38]. The ligand–protein complex was stabilized by hydrophobic interactions, and Thr135 was shown to be a key residue affecting specificity (Figure 3A). The second compound, (1S,2R)-2-(4-allyl-2,6-dimethoxyphenoxy)-1-(3,4,5-trimethoxyphenyl,) established a hydrogen bond with Ser133 and bound to aldol reductase with an affinity of −9.8 kcal/mol (Figure 3B). The third compound, (S)-5-Allyl-2-((1-(3,4-dimethoxyphenyl)propan-2-yl)oxy), showed significant hydrophobic contacts with Asp134 but did not generate hydrogen bonds with aldol reductase (Figure 3C).

The interaction profile of these compounds could contain significant knowledge for the development of inhibitors that mainly use hydrophobic interactions, offering another method of aldol reductase inhibition. Aldol reductase’s hydrophobic active site pocket allows it to interact with nonpolar ligands. The GROMACS 2023 was used to further simulate the top-scoring docking models. The conformational stability and kinetics of the interaction over 200 ns were shown by the MD simulation of the Human Aldose Reductase complexed with (S)-5-Allyl-2-((1-(3,4-dimethoxyphenyl)propan-2-yl)oxy)-1,3-dimethoxybenzene. To assess the stability of the protein–ligand complex in comparison to the unbound protein, RMSD (root mean square deviation) and RMSF (root mean square fluctuation) trajectory analyses were performed. The MD simulation results demonstrated that (S)-5-Allyl-2-((1-(3,4-dimethoxyphenyl)propan-2-yl)oxy)-1,3-dimethoxybenzene stabilizes human aldose reductase. When compared to the unbound protein, the protein–ligand complex’s reduced RMSD and RMSF values lend credence to the idea that ligand binding improves structural integrity while reducing the conformational flexibility (Figure 4A,B). Because it signifies a longer association and greater efficacy of the ligand under physiological conditions, its stability is essential for the functional inhibition of aldose reductase.

The results provide a new approach for the treatment of diabetic neuropathy through the use of a polyherbal extract as an aldose reductase inhibitor. In contrast to conventional inhibitors such as epalrestat, which have inconsistent efficacy and undesirable effects, PHE could synergize together to reduce oxidative stress, inflammation, and hyperglycemic damage—the three main causative factors in the progression of diabetes mellitus through the combination of several bioactive compounds. The molecular docking studies have shown that the extract has a high binding affinity of −11.4 kcal/mol to inhibit aldose reductase, which indicates its potential to prevent oxidative stress caused by the activation of the polyol pathway, considered to be one of the key components of DN. The in silico results, however, require experimental confirmation. In vivo studies should be conducted to determine the extract’s efficacy, pharmacodynamics, and pharmacokinetics in biological systems. Absorption, distribution, metabolism, excretion, and toxicity (ADMET) profiling should also be performed to determine the safety and potential therapeutic application. This work points out a possibility of integrating conventional medicine knowledge with modern drug designing methods. Apart from its ability to provide a sustainable alternative to synthesized drugs, this approach will find support for DN management with innovation and tradition while working in tandem with global plant-based therapy. Future investigations should seek to confirm these results in vivo and in vitro, as well as carry out toxicity and pharmacokinetic analyses to assess the compounds’ potential for medicinal development. The development of a novel aldol reductase inhibitor is strongly supported by these findings. This study highlights the importance of research conducted using animal models, such as rats, in the future to validate predicted data and findings. In vivo model studies give insight into the pharmacodynamics, efficacy, and therapeutic benefits of PHE. This research accordingly helps in filling the gap between the in silico prediction and clinical application.

## 5. Conclusions

In order to search for the presence of phytochemicals that could be utilized in drug development, the current investigation used GC-MS analysis of ethanolic PHE composed of *Citrullus colocynthis* (Bitter Apple), *Curcuma longa* (Turmeric), and *Myristica fragrans* (Nutmeg). The findings demonstrated that major phytoconstituents of PHE have high anti-inflammatory, neuroprotective, antidiabetic, and antioxidant properties. Furthermore, the present work was supported with computational analysis. The present work studied the extracted phytoconstituent of these medicinal plants against the target aldose reductase protein, which plays key role in diabetes mellitus. The computational analysis shows the use of PHE in the treatment of diabetic neuropathy through aldose reductase inhibition. Among the top rank binding scores, 4-isopropyl-1,6-dimethyl-1,2,3,4-tetrahydronaphthalene shows the highest binding affinity with aldose reductase protein with −11.4 kcal/mol. The stability and interaction of top ranked compounds (4-isopropyl-1,6-dimethyl-1,2,3,4-tetrahydronaphthalene;1S,2R)-2-(4-allyl-2,6-dimethoxyphenoxy)-1-(3,4,5-trimethoxyphenyl)-1-propanol; (S)-5-Allyl-2-((1-(3,4-dimethoxyphenyl)propan-2-yl)oxy)-1,3-dimethoxybenzene) with aldol reductase were confirmed using the molecular docking.

The limitation of the current investigation is that it did not cover the in vivo and in vitro experiments to validate the predicated efficacy of the polyherbal extract to assess their therapeutic potential in diabetic neuropathy. By using structure–activity relationship (SAR) investigations to maximize their binding efficiency and specificity, these leads can be further refined.

## Figures and Tables

**Figure 1 cimb-47-00075-f001:**
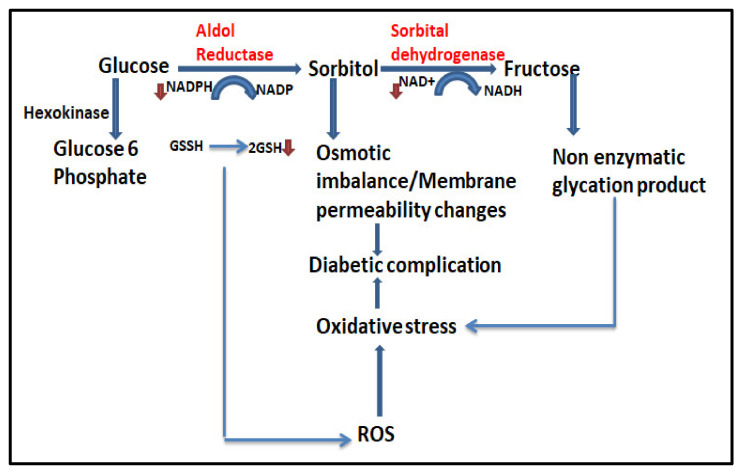
Schematic representation of polyol pathway of glucose metabolism, accumulation of sorbitol causes osmotic stress, which leads to the pathogenesis of diabetic complication.

**Figure 2 cimb-47-00075-f002:**
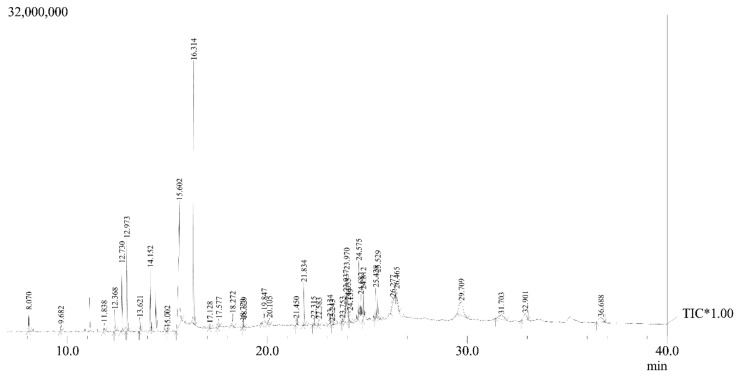
GC-MS chromatogram of PHE showing the presence of different compounds at different retention times. * Tentatively Identified Compounds (TIC).

**Figure 3 cimb-47-00075-f003:**
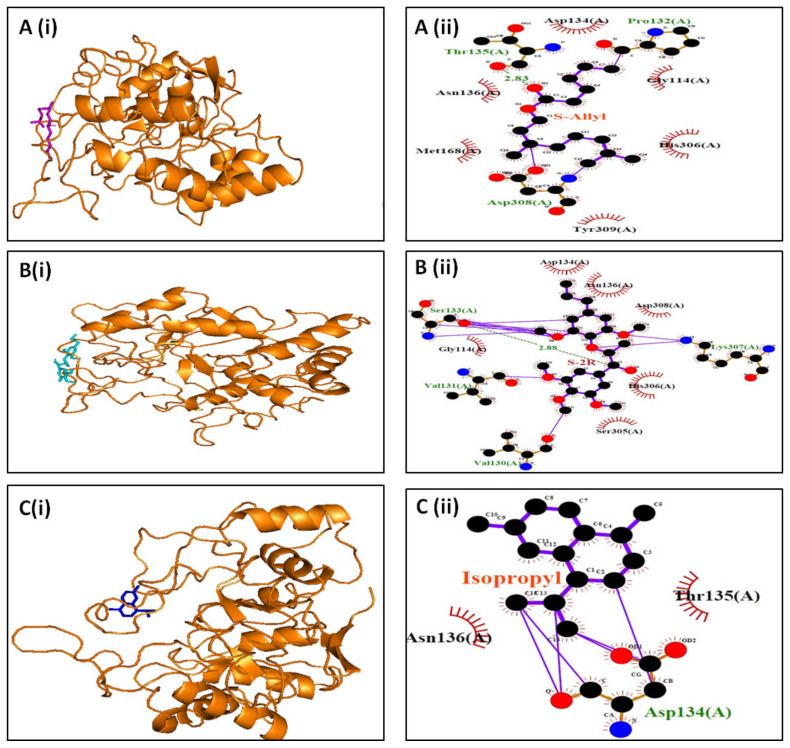
Presents a representative image illustrating the docking analysis of selected ligands with high binding affinity for aldol reductase. (i) The 3D interaction of the ligands with aldol reductase is depicted, with each ligand represented in stick form and shown in different colors: (**A**) (i) 4-isopropyl-1,6-dimethyl-1,2,3,4-tetrahydronaphthalene (in magenta); (**B**) (i) (1S,2R)-2-(4-allyl-2,6-dimethoxyphenoxy)-1-(3,4,5-trimethoxyphenyl)-1-propanol (in cyan); and (**C**) (i) (S)-5-Allyl-2-((1-(3,4-dimethoxyphenyl)propan-2-yl)oxy)-1,3-dimethoxybenzene (in blue). (**A**) (ii), (**B**) (ii), (**C**) (ii) The 2D interaction of the ligands with aldol reductase is generated by LigPlot v2.2.

**Figure 4 cimb-47-00075-f004:**
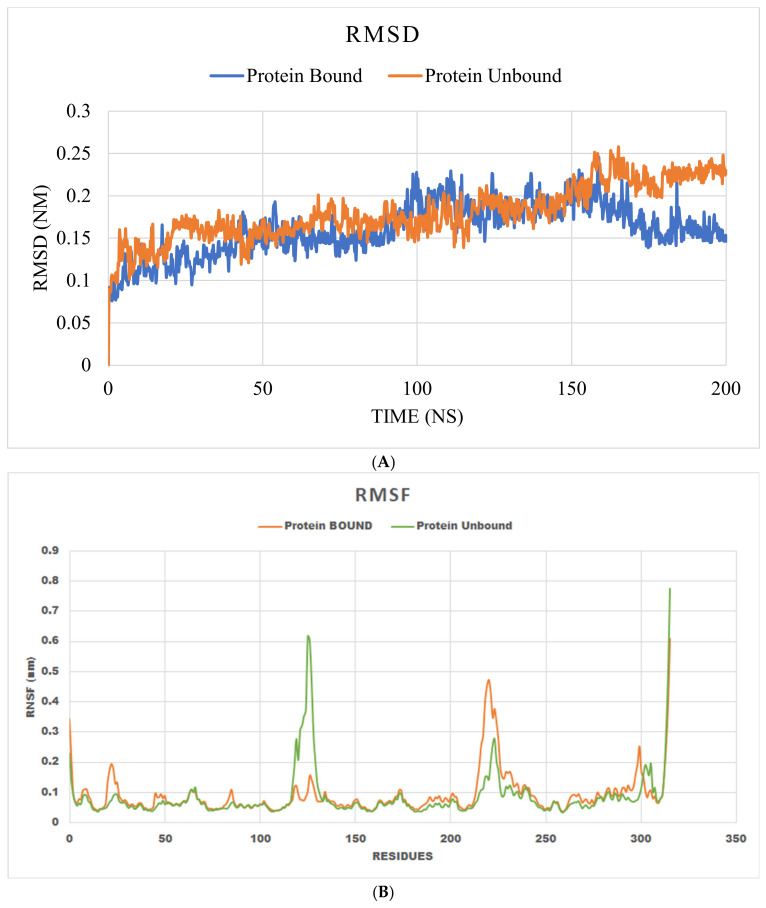
(**A**) Represents the RMSD graph of unbound protein (blue color) and protein-bound ligand (red color) over 200 ns of simulation. (**B**) Represents the RMSF graph of unbound protein (red color) and protein-bound ligand complex (green color) over 200 ns of simulation.

**Table 1 cimb-47-00075-t001:** Plant and its part used in preparation of the polyherbal extract.

S. No.	Botanical Name	English Common Name	Indian Common Name (Hindi)	Part Used
1.	*Citrullus colocynthis* L.	Bitter Apple	Indrayana	Fruit
2.	*Curcuma longa* L.	Turmeric	Haldi	Rhizome
3.	*Myristica fragrans* Houtt	Nutmeg	Jaiphal	Nut
4.	*Myristica fragrans* Houtt	Nutmeg	Jawitri	Aril

**Table 2 cimb-47-00075-t002:** Bioactive compounds present in PHE (GCMS analysis).

Peak	Name	RT (min)	Area%	MF	MW (g/mol)	Classification of Compounds
1	3-Cyclohexen-1-ol, 4-methyl-1-(1-methylethyl)-, (R)-	8.070	1.23	C_10_H_18_O	154	Alcohol
2	1,3-Benzodioxole, 5-(2-propenyl)-	9.682	0.56	C_10_H_10_O_2_	162	Phenylpropanoid
3	Phenol, 2-methoxy-4-(1-propenyl)-, (Z)-	11.838	0.76	C_10_H_12_O_2_	164	Alcohol
4	1,2-dimethoxy-4-[(1e)-1-propenyl]benzene	12.368	1.43	C_11_H_14_O_2_	178	Phenylpropanoid
5	1,3-Benzodioxole, 4-methoxy-6-(2-propenyl)-1,3 benzodioxole	12.730	4.45	C_11_H_12_O_3_	192	Phenylpropanoids
6	Benzene, 1,2,3-trimethoxy-5-(2-propenyl)-	12.973	5.53	C_12_H_16_O_3_	208	Phenylpropanoids
7	Phenol, 2,6-dimethoxy-4-(2-propenyl)-	13.621	0.97	C_11_H_14_O_3_	194	Phenylpropene
8	Dodecanoic acid, TMS derivative	14.152	3.84	C_15_H_32_O_2_Si	272	Fatty acid
9	Methyl tetradecanoate	15.002	0.20	C_15_H_30_O_2_	242	Fatty acid methyl ester
10	Tetradecanoic acid	15.602	20.38	C_14_H_28_O_2_	228	Fatty acid
11	Myristic acid, TMS derivative	16.314	17.95	C_17_H_36_O_2_Si	300	Fatty acid
12	Hexadecanoic acid, methyl ester	17.128	0.27	C_17_H_34_O_2_	270	Fatty acid methyl ester
13	n-Hexadecanoic acid	17.577	0.72	C_16_H_32_O_2_	256	Fatty acid
14	Carbonic acid, monoamide, N-octadecyl-, decyl ester	18.272	0.99	C_29_H_59_NO_2_	453	Ester
15	9,12-Octadecadienoic acid, methyl ester	18.770	0.35	C_19_H_34_O_2_	294	Fatty acid methyl ester
16	9-Octadecenoic acid, methyl ester, (9E)-	18.829	0.43	C_19_H_36_O_2_	296	Fatty acid methyl ester
17	9-Tetradecenoic acid, (E)-, TMS derivative	19.847	1.80	C_17_H_34_O_2_Si	298	Fatty acid
18	Cyclohexanol, 1-methyl-4-(1-methylethylidene)-, acetate	20.105	0.68	C_12_H_20_O_2_	196	Terpene
19	Isobutyl 2-(4-methylcyclohex-3-enyl)propan-2-yl carbonate	21.450	0.38	C_15_H_26_O_3_	254	Terpene
20	p-Menth-8-en-3-ol, acetate	21.834	2.54	C_12_H_20_O_2_	196	Terpene
21	3,7-Dimethyloct-6-en-1-yl tetradecanoate	22.315	0.51	C_24_H_46_O_2_	366	Fatty acid
22	(2E)-3,7-dimethyl-2,6-octadienyl hexanoate	22.583	0.36	C_16_H_28_O_2_	252	Fatty acid
23	1-Propanone, 1-(2,4-dimethoxyphenyl)-	23.134	0.34	C_11_H_14_O_3_	194	Ketone
24	(S)-5-Allyl-2-((1-(3,4-dimethoxyphenyl)propan-2-yl)oxy)-1,3 dimethoxybenzene	23.243	0.23	C_22_H_28_O_5_	372	Ether
25	4-isopropyl-1,6-dimethyl-1,2,3,4-tetrahydronaphthalene	23.753	0.20	C_15_H_22_	202	Hydrocarbon
26	Licarin A	23.937	0.55	C_20_H_22_O_4_	326	Neolignan
27	Licarin B	23.970	1.53	C_20_H_20_O_4_	324	Neolignan
28	(S)-5-Allyl-1,3-dimethoxy-2-((1-(3,4,5-trimethoxyphenyl)propan-2yloxy) benzene	24.035	0.62	C_23_H_30_O_6_	402	Ether
29	Phenol, 2,2′-methylenebis[6-methoxy-3-(2-propenyl	24.139	0.64	C_21_H_24_O_4_	340	Alcohol
30	(1S,2R)-2-(4-Allyl-2,6-dimethoxyphenoxy)-1-(3,4-dimethoxyphenyl)propyl acetate	24.575	3.27	C_24_H_30_O_7_	430	Phenylpropanoids
31	6-Methoxyeugenyl isovalerate	24.682	0.88	C_16_H_22_O_4_	278	Fatty acid
32	(1S,2R)-2-(4-Allyl-2,6-dimethoxyphenoxy)-1-(3,4-dimethoxyphenyl)propan-1-ol	24.812	1.68	C_22_H_28_O_6_	388	Phenylpropanoids
33	1-Phosphacyclopent-2-ene, 1,2,3-triphenyl-5-dimethylmethylene	25.428	1.95	C_25_H_23_P	354	Hydrocarbon
34	(1S,2R)-2-(4-allyl-2,6-Dimethoxyphenoxy)-1-(3,4,5-trimethoxyphenyl) propan-1-ol-rel-	25.529	3.80	C_23_H_30_O_7_	418	Phenylpropanoids
35	Tetracosapentaene, 2,6,10,15,19,23-hexamethyl	29.709	8.71	C_30_H_52_	412	Triterpenoids
36	Tetracosenoic acid, 2-[(trimethylsilyl)oxy	26.465	1.47	C_28_H_56_O_3_Si	468	Ester
37	Cholesta-5,20-dien-3-ol, (3.beta.)-ol	31.703	2.76	C_27_H_44_O	384	Sterol
38	Adipic acid, 2,4-dimethylpent-3-yl eicosyl ester	32.901	2.70	C_33_H_64_O_4_	524	Ester
39	Hexadecanoic acid, 2-hydroxy-1,3-propanediyl ester	36.688	2.37	C_35_H_68_O_5_	568	Fatty acid

**Table 3 cimb-47-00075-t003:** List of the selected compounds after screening follows the Lipinski rule and shows the binding affinity with the target aldol reductase protein.

S. No.	Compounds	Binding Affinity (kcal/mol)
1.	(S)_5_Allyl_1_3_dimethoxy_2_((1_(3_4_5_trimethoxyphenyl)	−9.3
2.	1-Propanone, 1-(2,4-dimethoxyphenyl)	−9.5
3.	3,7-Dimethyloct-6-en-1-yl tetradecanoate	−9.7
4.	4-isopropyl-1,6-dimethyl-1,2,3,4-tetrahydronaphthalene	−11.4
5.	(1S,2R)-2-(4-allyl-2,6-Dimethoxyphenoxy)-1-(3,4,5-trimeth	−9.8
6.	Benzene, 1,2,3-trimethoxy-5-(2-propenyl)-	−8.0
7.	3-Cyclohexen-1-ol, 4-methyl-1-(1-methylethyl)-, (R)-	−9.5
8.	Isobutyl 2-(4-methylcyclohex-3-enyl)propan-2-yl carbonate	−8.6
9.	(S)-5-Allyl-2-((1-(3,4-dimethoxyphenyl)propan-2-yl)oxy)	−10.3
10.	p-Menth-8-en-3-ol, acetate	−9.1
11.	TETRACOSAPENTAENE, 2,6,10,15,19,23-HEXAMETHY	−8.3

## Data Availability

Data are contained within the article and Appendix A.

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
