# Peer review of "GC-MS Profiling of Ethanol-Extracted Polyherbal Compounds from Medicinal Plant (Citrullus colocynthis, Curcuma longa, and Myristica fragrans): In Silico and Analytical Insights into Diabetic Neuropathy Therapy via Targeting the Aldose Reductase"

_cimb, 2025, doi:10.3390/cimb47020075_

Round 1

Reviewer 1 Report

Comments and Suggestions for Authors

The authors describe the therapeutic effects of a polyherbal extract from Citrullus colocynthis, Curcuma longa, and Myristica fragrans using in silico and analytical methods for diabetic neuropathy. The authors claimed high affinity binding to aldose reductase, an enzyme involved in the pathophysiology of diabetic neuropathy through computational testing. The main findings in this manuscript are limited to the GC-MS profiling of the studied plants. The manuscript uses appropriate language.

This work merits publication after major corrections.

1.       The title of the manuscript should be modified to reflect the main research idea and results. Words like pharmacokinetic evaluation and potential against diabetic neuropathy are out of context.

2.       The abstract should contain the main results of the research, what compounds were identified, and which ones are the most active in the computational studies and their results. A big proportion of the abstract discusses the potential of future research instead. The research still needs both in vitro and in vivo studies to claim the proposed mode of action.

3.       The authors didn’t mention whether the compounds identified from the extract are novel or already reported for the studied plants. This needs to be well clarified since it is the main research finding.

4.       The authors used an ethanol extract. This is an important piece of information that should be included in the abstract and may be the title and a literature review about other types of extract or previous ethanol extract should be conducted.

5.       The authors did not clarify on what basis they chose the targets and why they expect potential activity against diabetic neuropathy. Are there compounds with similar structures reported for the same activity or did they perform target prediction in a website or program.

6.       The authors should redesign the presentation of their manuscript based on identification and profiling of compounds from natural products extract. The profiled compounds were then screened against potential target that needs to be rationally identified and finally this target activity could be correlated with the claimed disease for future biological investigations.

7.       The results part especially for the GC-MS should enriched and emphasized as it is the main research findings.

8.       The pharmacokinetic prediction should have a separate part not under molecular docking and moved to the end after MD simulation.

9.       The conclusion part should be rewritten based on the previous comments.

Reviewer 2 Report

Comments and Suggestions for Authors

Comments to the Authors,

I read with great interest this manuscript ID cimb-3389720 entitled “GC-MS Profiling, MD Simulation, and Pharmacokinetic Evaluation of the Polyherbal Extract with Potential Against Diabetic Neuropathy”. This is an interesting study. Attached below my comments that I hope would add to the manuscript.

Title:

The title is not representative of the polyherbal extract used. It would be nice to be more specific and add the plant extracts used.

Introduction: 

It would be nice to replace the term diabetic individuals by people with diabetes.

Methodology:

It would be nice to add the exact technique of preparation and the concentration of each extract in grams and how they were extracted and purified.

What about the safety issues regarding this extract?

How will it be prepared.

Discussion:

·       It would be nice to replace the term we and our by the current study throughout the manuscript.

·      The discussion is rather a narration of the results, you should discuss your findings in view of available literature.

It would be nice to add a limitations section

Reviewer 3 Report

Comments and Suggestions for Authors

In this study, Kausar and colleagues identified the major compounds of an herbal extract composed of nutmeg, bitter apple and turmeric using GC-MS. Then they screened the bioactive compounds with potential to bind aldol reductase (AR), a key enzyme in the polyol pathway which plays a major role in the pathogenesis of diabetes mellitus including diabetic neuropathy. The molecular docking studies allowed them to identify three compounds with the highest binding activity to AR, suggesting that their pharmacological and safety profile should be investigated in the future.

Overall, the study seems well conducted and the manuscript well-written. The only major issue this reviewer raises deals with the tone of the text – the authors claim they have investigated the therapeutic effects of the herbal extract in the inhibition of aldol reductase, which seems slightly misleading. Even though that seems to be their goal in the long-term, the authors simply screened several compounds for their binding activity of AR in silico, while no demonstration of a beneficial (i.e., therapeutic) effect was made. As such, this reviewer recommends that several key sentences in the manuscript be changed in order to provide a more concise description of the aims and results of the study.

This reviewer also points out several minor issues:

A) Introduction

-            The term “lethal” should be replaced, as neuropathy is not lethal in itself, although it increases the risk of potentially lethal cardiovascular, metabolic and infectious events in diabetic patients;

-            The terms “tingling, numbness and excessive weakness” should be replaced with “paresthesia and weakness”;

-            Since the polyol pathway is a key player in diabetes, the authors would significantly improve the manuscript by providing a simple schematic of that pathway and of its negative cellular effects for the tissues/organs in diabetes;

-            What is meant by “traditional pharmacological therapies” when discussing isolated compounds such as zenerastat?;

-            The adverse effects or zenerastat and similar drugs are repeated in two continuous sentences. This should be simplified;

-            The term “holistic” does not seem to be the most appropriate when referring to potentially synergistic roles for herbal extracts. This should be revised;

-            The common names of the herbs should be presented along with their complete Latin denomination (the name of taxonomist is missing in two herbs);

B) Materials and Methods

-            Can the authors provide more details regarding the harvesting conditions of the herbs featured in the extract?;

-            The first sentence of this section suggests that the authors are not sure whether these herbs are used by local populations as a traditional remedy. This should be revised;

 C)        Results

Table 2 – units for RT and MW should be provided. Instead of “nature” of the compound perhaps it should be “Classification”?. Also, why are the names of some compounds written in capital letters? The names of many also seem to be incomplete;

Figure 1 should be improved, as the arrows and RT values are close together, decreasing readability of the graph;

D) Discussion

In the first sentence, where it reads “DN is a condition associated with diabetes” perhaps it could be better as “DN is a condition caused by diabetes”.

This reviewer disagrees with the sentence where the authors mention that current therapeutic methods are not curative for the root cause. Although they are not curative – as aldol reductase inhibition is not expected to be – they address the root mechanisms of the disease;

If the authors wish to claim that the compounds which show inhibition of AR are also involved in other pathways of diabetes, they need to stress this and provide references. Only then can they claim a multi-targeted approach for these compounds;

Round 2

Reviewer 1 Report

Comments and Suggestions for Authors

The authors have now performed the required modifications, and the manuscript is suitable for publication in its present form.